# Shorter Chitin Nanofibrils Enhance Pickering Emulsion Stability: Role of Length and Interfacial Network

**DOI:** 10.3390/foods15010076

**Published:** 2025-12-26

**Authors:** Qiyue Yang, Congying Chen, Xiaoyi Luo, Ruoxin Li, Zhenjun Zhu, Yehui Zhang, Xinglong Xiao, Wenjuan Jiao

**Affiliations:** 1School of Food Science and Engineering, South China University of Technology, Guangzhou 510640, China; yqy51522@163.com (Q.Y.); c1030514976@163.com (C.C.); 2Key Laboratory of Functional Foods, Ministry of Agriculture and Rural Affairs, Guangdong Key Laboratory of Agricultural Products Processing, Sericulture & Agri-Food Research Institute Guangdong Academy of Agricultural Sciences, Guangzhou 510610, China; 18739896783@163.com (X.L.); 13131171362@163.com (R.L.); zhangyehui@gdaas.cn (Y.Z.); 3Department of Food Science and Engineering, College of Life Science and Technology, Jinan University, Guangzhou 510632, China; zzj1904@jnu.edu.cn

**Keywords:** length-dependent, structure-activity, emulsion stability, crab shells

## Abstract

The structure–property relationship of chitin nanofibrils (NCh) with tailored lengths (L-, M-, S-NCh) and their efficacy in stabilizing Pickering emulsions were systematically investigated. The nanofibrils, produced via high-pressure homogenization and ultrasonication (20 or 60 min), were characterized by transmission electron microscopy (TEM). Emulsion stability was predominantly governed by nanofibril length and concentration, with S-NCh (shortest) exhibiting superior performance, as evidenced by its minimal creaming index, smallest droplet size (1.18 μm at 0.5%), and homogeneous microstructure observed by confocal laser scanning microscopy (CLSM). A critical stabilizer concentration of 0.05% was identified, below which instability occurred due to insufficient interfacial coverage. Rheological analysis confirmed shear-thinning behavior and solid-like viscoelasticity at high frequencies. CLSM microstructural observations directly confirmed nanofibril adsorption at the interface and the formation of a continuous network between droplets, elucidating the stabilization mechanism. These findings demonstrate that shorter chitin nanofibrils provide a marked improvement in emulsion stability, offering a superior biomass-derived alternative for the design of stabilizers in food and pharmaceutical applications.

## 1. Introduction

Pickering emulsions, first described by the British scientist S. U. Pickering in 1907 [1], represent a unique class of emulsified systems stabilized by micro- and nano-scale solid particles instead of traditional surfactants. These particles adsorb irreversibly at the oil–water interface, forming a rigid physical barrier that effectively prevents droplet coalescence and imparts exceptional stability against environmental stresses such as temperature and pH fluctuations [2]. Due to these advantages and the generally low toxicity of the solid particles, Pickering emulsions have garnered significant interest across diverse fields including food, cosmetics, pharmaceutical delivery, and materials science [3,4,5,6]. The stability of Pickering emulsions, type (oil-in-water, O/W, or water-in-oil, W/O), and functionality are predominantly governed by the characteristics of the stabilizing particles, such as their wettability, size, shape, and concentration [7]. While a wide range of solid particles, from synthetic polymers to natural biopolymers, have been explored [8], many natural options like alginate or protein particles often require chemical modification or complex processing to achieve optimal performance [9,10,11]. In this context, chitin emerges as a highly promising natural stabilizer. As a naturally abundant nitrogen-containing polysaccharide found in fungal cell walls and arthropod exoskeletons [12], chitin possesses an inherent structural robustness and interfacial activity. Its nanofibrils can be prepared via top-down (e.g., disintegration of bulk chitin) or bottom-up (e.g., electrospinning, self-assembly) approaches [13,14,15,16], offering flexibility in material design.

Nanochitin boasts numerous natural advantages, including non-toxicity, excellent biocompatibility, biodegradability, and antibacterial properties [17,18,19,20]. These attributes make it an ideal stabilizer for Pickering emulsions [21], with demonstrated potential in delivery systems, food packaging, and modulating lipid digestion [22,23,24]. For instance, studies have shown that chitin-stabilized emulsions can reduce lipid digestion rates by 30% compared to those stabilized by small-molecule surfactants [23], highlighting their utility in designing functional foods.

Despite this promising outlook, a clear structure–property relationship governing the efficacy of chitin nanoparticles in Pickering emulsions remains inadequately established. The correspondence between the basic structural parameters of chitin nanocrystals and their emulsifying functionality, as well as the emulsifying role and multifunctional properties of partially deacetylated chitin, have not been clearly elucidated. Moreover, the mechanisms by which different degrees of structural modification influence emulsion stability remain incompletely uncovered [25,26]. Even for chitin nanomaterials subjected to specific modifications such as enzymatic hydrolysis or acetylation, a systematic understanding of the correlation between structural parameters including aspect ratio, surface charge, and crystallinity and emulsion stability is still lacking. Furthermore, the quantitative impact of surface functionalization on their rheological and self-assembly behaviors in emulsions has not been established [27,28]. This study systematically investigated the stabilization mechanism of Pickering emulsions using chitin nanofibrils (NChs) with varying morphologies. We aimed to elucidate the correlation between the structural characteristics (length and concentration) of NChs and the resulting emulsion properties. A comprehensive analysis was conducted to evaluate the effects of NChs on storage stability, droplet size distribution, microstructure, and interfacial rheology. The findings provide deeper insights into the stabilization mechanism of NChs and confirm their potential as tailored, high-performance stabilizers for the design of advanced functional foods.

## 2. Materials and Methods

### 2.1. Material

Crab shells were purchased from Hongxin Aquatic Products Co., Ltd. (Zhanjiang, China). Azobisisobutyronitrile (AIBN, 98% purity), hydrochloric acid, sodium hydroxide, sodium hypochlorite, acetic acid, ethanol, phenethyl ether, and nitrogen were purchased from Guangzhou Qianhui Glass Instrument Co., Ltd. (Guangzhou, China). Corn oil (unsaturated fatty acid >80%) was purchased from Yihai Jiali Food Marketing Co., Ltd. (Shanghai, China). Nile red and Calcofluor white stain were purchased from Sigma Aldrich (Shanghai, China). All chemical reagents used were analytical-grade.

### 2.2. Preparation of Chitin Nanofibrils (NCh)

Chitin extraction was conducted from crab shells following a previous study with slight modification [13,15,29,30]. Briefly, the crab shells were immersed in 1 M HCl (24 h) and 1 M NaOH solutions (24 h) with 3 cycles, to remove minerals (e.g., calcium) and proteins. Then, the processed crab shells were decolorized with NaClO_2_ solution (0.5 wt%, pH 5.0 with acetic acid) at 70 °C for at least 2 h until complete whitening. The purified chitin shells after decolorization were thoroughly rinsed with distilled water, followed by mechanically disintegration using a grinder (BJ-300A, Deqing Baijie Electric Appliance Co., Huzhou, China). The moisture content of the obtained chitin was determined by an analyzer (DHS-16A, Jinghai Instrument Co., Shanghai, China)

The purified chitin was deacetylated with NaOH solution (33 wt %) at 90 °C for 3.5 h with a solid-to-liquid ratio of 0.04 g/mL. Then, the partially deacetylated chitin was fully washed with water until reaching a pH of 7, followed by an overnight dialysis to further remove impurities. Before mechanical nanofibrillation, the partially deacetylated chitin was dispersed in water (0.5 wt% concentration) and the pH was adjusted to 3 using acetic acid under continuous stirring. The partially deacetylated chitin suspension was homogenized by a high-speed blender (T18, IKA (Guangzhou) Instrument Equipment Co., Ltd., Guangzhou, China) at 10,000 rpm for 5 min in an ice-water bath and then maintained at 4 °C.

In this study, the finer chitin nanofibrils were obtained by mechanical disintegration, such as sonication and microfluidizer. One kind of the finer chitin nanofibrils was carried out with a microfluidizer (M-110EH-30, MFIC, Weston, MA, USA) with a pressure of 23,500 psi and four cycles, which was named as L-NCh. The finer chitin nanofibrils (M-NCh and S-NCh) were generated through sonication (VCX500, SONICS, Newtown, CT, USA) at 300 W for 20 min and 60 min (5 s sonication with a 2 s interval), respectively. Finally, all samples were centrifuged to remove aggregates and the supernatant was collected. And nanofibrils suspensions were standardized to 0.5 wt% concentration through dilution (distilled water, pH 3) or rotary evaporation.

### 2.3. Preparation of Pickering Emulsions

Pickering emulsions were prepared using either corn oil (20 wt%) or the different kinds of Nch suspensions (L-, M-, S-NCh) as stabilizer (80 wt%). The NCh suspensions were diluted to specific concentration (0.5%, 0.3%, 0.2%, 0.1%, 0.05%, 0.01%, 0.005%, and 0.001%) with MQ water at pH 3.0 (acetic acid) to ensure sufficient protonation of NCh [31,32]. The oil phase and NCh suspensions were mixed to prepare the coarse emulsions using a high-speed blender (T18, IKA (Guangzhou) Instrument Equipment Co., Ltd., Guangzhou, China) at 10,000 rpm for 4 min, followed by sonication (VCX500, SONICS, Newtown, CT, USA) at 300 W for 2 min (3 s sonication with a 2 s interval). The two-step emulsification process was carried out in an ice bath to avoid overheating.

### 2.4. Characterization of Chitin Nanofibrils

#### 2.4.1. Surface Characteristics

The morphology of the NCh was observed by a transmission electron microscope (TEM; JEM-1400Plus, Jeol, Tokyo, Japan). Deionized water was used to dilute the NCh suspensions to a concentration of 0.005%. TEM specimens were prepared by drop-casting method on an ultrathin carbon film supported by a carbon film on a copper grid. The sample was then negatively stained with uranyl acetate solution prior to being air-dried at room temperature. Microscopic observations were carried out under an acceleration voltage of 120 kV [33].

#### 2.4.2. Deacetylation Degree (DD) of Chitin

The DD of NCh suspensions was determined by a conductivity meter (DDS-11A, Shanghai Yidian Scientific Instruments Co., Shanghai, China). The NCh suspensions (2 g dry matter content) were supplemented with 5 μL of 5 M KCl solution to enhance ionic strength for improved buffering capacity. A 30 mL of 0.1 M hydrochloric acid (HCl) standard solution was added into the reaction system, followed by progressive titration with 0.01 M sodium hydroxide (NaOH) standard solution. pH and conductivity values were recorded following each 0.5 mL addition of NaOH solution. The DD is calculated based on the inflection point of the conductivity titration curve. The mass fraction of the DD is determined using the following formula.(1)$$w_{1}=\frac{∆V\times c_{1}\times 10^{-3}\times 16}{m_{1}\times 0.0994}\times 100\%$$

Among them, ∆*V* represents the volume of NaOH standard titration solution consumed between two “jump” points, in milliliters, *c*_1_ represents the concentration of NaOH standard titration solution, measured in moles per liter, 10^−3^ is the unit conversion factor, 16 is the molar mass of the amino group, measured in grams per mole (g/mol), *m*_1_ is the sample mass measured in grams, and 0.0994 is the theoretical amino content.

#### 2.4.3. Functional Group Change Analysis

Infrared spectra are obtained through a Fourier Transform Infrared (FTIR) spectrometer (VERTEX, Bruker, Billerica, MA, USA) to analyze the changes in functional groups of materials during processing. The spectrometer was set at a resolution of 4 cm^−1^ and a wavelength range of 4000–400 cm^−1^. Crab shell powder, chitin, and NCh (2 mg each) were ground into fine powders. Subsequently, these powders were mixed with KBr at a ratio of 1:100, and the mixtures were compressed into tablets for spectral analysis.

#### 2.4.4. Crystallinity

The crystallinity of the sample was determined using an X-ray diffractometer (X’pert Powder, Panaco, Auckland, The Netherlands). The instrument was operated with a Cu Kα radiation parameter of λ = 0.1541 nm, a voltage of 40 kV, and a filament current of 30 mA. The scanning rate was set at 1 °/min and the angular range of 2θ (diffraction angle) spanned from 5° to 60°.

The crystallinity index value was calculated according to the following formula [34].(2)$$CrI\%=\frac{I_{110}-I_{am}}{I_{110}}\times 100$$
where *I*_110_ represents the maximum intensity of diffraction at 2θ = 19° and *I_am_* represents the intensity of the amorphous portion (2θ = 12.6°).

### 2.5. Characterization of Pickering Emulsions

#### 2.5.1. Creaming Index(CI)

The CI of emulsions was studied by placing the emulsions in a sealed glass tube and storing it at 4 °C for 7 days. The height of the emulsion serum layer was measured at scheduled time points.

The creaming index was calculated according to the following formula.(3)$$CI=\frac{H_{S}}{H_{T}}\times 100\%$$
where *H_S_* is the height of the lower serum layer after storage and *H_T_* is the total height of the emulsion, in cm.

#### 2.5.2. Particle Size and Zeta Potential

The particle size and zeta potential of the Pickering emulsions were determined using a Zetasizer Nano ZS90 (Malvern Instruments Ltd., Worcestershire, UK). The refractive indices of the corn oil and the aqueous phase were set at 1.471 and 1.33, respectively. Before analysis, the emulsions were diluted with ultrapure water to 0.01%. The calculation formula for the particle size is as follows [35]:(4)$$D_{\left(4,3\right)}=\frac{\sum n_{i}d_{i}^{4}}{\sum n_{i}d_{i}^{3}}$$
where *d_i_* is the diameter of the droplet with a fraction of *i*, and *n_i_* is the number of droplets with a size of *d_i_*.

#### 2.5.3. Microstructure Measurements

A CLSM (LSM710, Carl Zeiss, Oberkochen, Germany) was employed to characterize the microstructure of emulsion droplets at initial preparation (Day 1) and post-7-day storage. Oil-phase labeling was achieved by blending the emulsion with 0.1% (*w*/*v*) Nile Red ethanol solution at a 50:1 (*v*/*v*) ratio, while NCh was fluorescently tagged through combination with 0.1% (*w/v*) Calcofluor White solution at a 20:1 (*v*/*v*) ratio. Stained specimens were equilibrated at 25 °C for 10 min prior to analysis. Microscopic observation was performed by depositing 6 μL aliquots onto glass slides under coverslip-mounted conditions. Fluorescence detection parameters were standardized with excitation/emission wavelengths set at 488/539 nm for Nile Red and 365/435 nm for Calcofluor White, respectively. The fused fluorescence images were processed using ZEN 2010 software.

#### 2.5.4. Rheological Properties

The apparent viscosity of emulsions was determined using an rheometer (HR-1, TA Instruments, New Castle, DE, USA) equipped with a parallel plate geometry (PP25, 0.5 mm gap). Samples underwent pre-shearing conditioning at 10 s^−1^ for 60 s. Rheological characterization was performed through controlled shear rate ramping (0.01–100 s^−1^) while monitoring the corresponding viscosity profile. Flow behavior was analyzed using the Herschel–Bulkley model, expressed as(5)$$\tau =\tau_{0}+k\gamma^{n}$$
where *τ* and *τ*_0_ are the shear stress and yield stress, Pa. *k* is the consistency, and Pa·s^n^. *n* is the traffic index.

Dynamic viscoelastic properties were characterized using the same rheometer configuration at 25 °C. Strain amplitude sweeps were performed at a fixed angular frequency of 10 rad/s (0.01–100% strain) to determine the linear viscoelastic region (LVR). Subsequently, frequency sweeps were conducted within the identified LVR (1.0% constant strain) over an angular frequency range of 0.1–100 rad/s. Storage modulus (G′) and loss modulus (G″) were recorded as functions of frequency to construct the dynamic mechanical spectra.

### 2.6. Statistical Analysis

All experiments were conducted with a minimum of three independent replicates (n ≥ 3). These replicates were performed on separately prepared batches of nanofibrils and emulsions at different times to assess inter-batch reproducibility. Results are presented as mean ± standard deviation (SD). Statistical significance (*p* < 0.05) was determined by one-way analysis of variance (ANOVA) followed by Duncan’s multiple range test, using SPSS software (version 27; IBM, Armonk, NY, USA). Graphs were generated with GraphPad Prism (version 8.0.2) software. Key measurements, such as laser diffraction for droplet size and rheological analysis, were conducted following standardized protocols with calibrated instruments to ensure consistency across replicates.

## 3. Results

### 3.1. Deacetylation Degree (DD) of Chitin

The deacetylated sample was classified as chitin based on its degree of deacetylation (DD), which was determined to be 21.25% by conductometric titration. This value is well below the 85% threshold required for chitosan classification [36]. The titration profile (Figure 1A) exhibited three characteristic phases: a steep initial slope corresponding to the neutralization of residual HCl, a subsequent plateau indicating the deprotonation of free amino groups, and a final sharp increase due to excess NaOH.

### 3.2. Surface Characteristics

Transmission electron microscopy (TEM) was employed to analyze the morphological characteristics of diluted nanofibrils suspensions. The micrographs at three magnifications (2 μm, 500 nm, and 200 nm) clearly show that all variants (L-, M-, S-NCh) were well-dispersed and exhibited a rod-like morphology. A progressive decrease in length from L-NCh to S-NCh was observed (Figure 2).

### 3.3. Functional Group Change

Fourier Transform Infrared spectroscopy (FTIR) was employed to investigate functional group changes in five different samples: raw crab shell powder, deacetylated chitin, and three lengths of NChs (L-, M-, S-NCh). As shown in the FTIR spectra (Figure 1C), characteristic absorption bands of chitin were observed, including amide I (1655 cm^−1^ and 1620 cm^−1^) and amide II (1560 cm^−1^) [37], which confirmed the presence of amide, hydroxyl, methyl, and carbonyl functional groups. The C-H stretching vibrations in the 2800–2900 cm^−1^ range were assigned to methyl and methylene groups, while the broad O-H stretching band around 3450 cm^−1^ was related to varying degrees of sample hydration. Notably, no correlation was found between the intensity of the hydroxyl band and the primary properties of the chitin material.

### 3.4. Crystallinity

X-ray diffraction (XRD) analysis was conducted on five samples (crab shell, chitin, L-NCh, M-NCh, and S-NCh) following the classification methodology described in Section 3.3. The resulting diffraction patterns (Figure 1B) exhibited characteristic chitin peaks at 2θ= 9.3°, 12.7°, 19.3°, 23.1°, and 26.6° [34]. Crystallinity values were quantitatively evaluated based on the two prominent peaks at 2θ = 12.7°and 19.3°, yielding 77.5% for crab shell, 79.4% for chitin, 78.7% for L-NCh, 79.9% for M-NCh, and 75.4% for S-NCh. An increase in crystallinity was observed after the deacetylation process, which can be attributed to homogenization and short-term ultrasonic treatment. The amorphous regions of chitin were disrupted, exposing and enhancing the relative content of crystalline regions [38]. In contrast, a reduction in crystallinity occurred under prolonged ultrasonication (60 min) or high-pressure microfluidization, indicating the disruption of the crystalline structure due to intensive physical treatment. This trend indicates that the intensive mechanical processing necessary to produce the shortest nanofibrils (S-NCh) not only reduced their length but also partially disrupted the crystalline order, thereby increasing the relative proportion of amorphous domains. Such increased amorphous content is frequently linked to enhanced surface activity and defect-rich architectures, features that are acknowledged in advanced amorphous nanomaterials for promoting interfacial interactions and functionality [39]. Consequently, the moderately lower crystallinity of S-NCh may contribute to its superior interfacial performance by improving nanofibril flexibility [40], which allows for more effective conformation and denser packing at the curved oil–water interface, while also potentially increasing the accessibility of surface groups for irreversible adsorption. In contrast, the higher crystallinity of L-NCh and M-NCh corresponds to their tendency to form entangled networks within the aqueous phase, a behavior typical of more rigid rod-like particles. Thus, although fibril length remains the dominant factor, the concurrent modulation of crystallinity during processing generates a synergistic effect, in which a moderate reduction in crystallinity, together with shortened length, facilitates a transition from a network-forming stabilizer to one with superior interfacial stabilization capability.

### 3.5. Creaming Index

The stability of NCh-stabilized emulsions was assessed by visual inspection and quantitative measurements. On the day of preparation and after seven days of storage at 4 °C, the Pickering emulsions were photographed in order of increasing concentration for comparative analysis (Figure 3). The serum layer thickness was measured on Day 7, and the results of the emulsion index are summarized in Table 1. Visible differences were observed among the emulsions prepared by L-, M-, and S-NCh. Partial delamination occurred in L-NCh formulations (0.001%, 0.005%, 0.01%) within one hour and serum layers were also detected in M-NCh systems at the same concentrations. In contrast, S-NCh formulations remained homogeneous, indicating superior stability. These observations were consistent with Day 7 results (Figure 3B,D,F) and creaming index data (Table 1).

A significant decrease in stability was observed at NCh concentration below 0.05%, which was attributed to inadequate nanofibril coverage at the oil–water interface. This insufficient coverage led to increased interfacial tension and droplet coalescence [41], compromising the formation of a stable interfacial film and thus reducing overall emulsion stability.

### 3.6. Particle Size and Zeta Potential

The droplet size was expressed as the volume moment mean diameter, D(4,3). This metric was chosen over percentile values (e.g., DV50) because it provides a volume-weighted average that is particularly sensitive to larger droplets within the distribution, offering a more relevant indicator for assessing emulsion stability and the risk of coalescence or creaming [42,43,44]. The average particle size and distribution of the Pickering emulsions stabilized by NCh were measured immediately after preparation (Day 1) and after seven days of storage (Day 7) (Figure 4). The zeta potentials of the emulsions are shown in Appendix A. Analysis revealed an exponential increase in particle size with decreasing NCh concentration, observed on both Day 1 and Day 7. At a concentration of 0.5%, all three nanofibrils (L-, M-, S-NCh) produced emulsions with minimal particle sizes of 3.80, 1.28, and 1.18 μm, respectively. As shown in Figure 4, emulsions prepared with nanofibril concentrations ≥ 0.05% maintained below 10 μm, indicating limited variation in size across this concentration range. This stability threshold suggests that the droplet interface becomes saturated with nanofibrils at a concentration above 0.05%, with any excess particles remaining in the continuous phase without significantly influencing emulsion droplet size [45].

The dense adsorption of shorter nanofibrils (S-NCh and M-NCh) with lower aspect ratios at the oil–water interface is conducive to more effective regulation of interfacial curvature and suppression of droplet coalescence [27]. In contrast, longer nanofibrils (L-NCh) with higher aspect ratios tend to aggregate in the continuous phase, impairing their ability to match the droplet interface curvature. Above 0.05% NCh concentration, interfacial saturation and excess particles promote a percolating network that restricts droplet mobility, thereby further enhancing stability.

At lower concentrations (0.01% and 0.005%), S-NCh emulsions maintained particle sizes below 10 μm with narrow size distributions, whereas M-NCh formulations exhibited broader distributions within the same range. A comparative assessment of the dispersibility revealed the hierarchy of effectiveness as S-NCh > M-NCh > L-NCh (Appendix A). In contrast to the emulsion (Day 1), the Day 7 measurements confirmed that emulsion stability improved with decreasing nanofibril length, aligning with the findings presented in Section 3.5.

Zeta potential analysis (Appendix A) revealed concentration-dependent behavior, characterized by an initial increase followed by a plateau, while maintaining temporal stability between Day 1 and Day 7 measurements. Emulsions stabilized by S-NCh and M-NCh were observed to maintain a more stable state throughout the seven-day storage period. As the pH conditions were strictly controlled during emulsion preparation, the variations in zeta potential primarily reflected a proportional relationship with chitin nanofibril concentration [46].

### 3.7. Microstructure Changes

CLSM was employed to obtain microscopic images of the emulsions, with the aqueous phase stained to appear fluorescent white and the oil phase labeled with Nile red. Figure 5 displays CLSM images of emulsions prepared with 0.01% chitin nanofibrils of different length on Day 7, whereas Figure 6 compares single- and dual- staining patterns at a concentration of 0.3%. The microscopic analysis validated the previously noted particle size trends, confirming that emulsion droplet size increased with decreasing stabilizer concentration and increasing nanofibril length—a phenomenon consistent with earlier reports [47,48]. Furthermore, Day 7 observations indicated pronounced droplet aggregation in emulsions stabilized with 0.05–0.5% L-NCh, in contrast to the homogeneously dispersed droplets observed in M-NCh and S-NCh formulations. Phase-selective fluorescent labeling revealed distinct microstructural features: chitin nanofibrils localized in the aqueous phase exhibited blue fluorescence, whereas the oil phase showed distinct red emission. Figure 6 demonstrates complete fluorescent coverage of the droplets, confirming adsorption of chitin at the oil–water interface and corroborating the proposed emulsion formation mechanism. In Figure 6, higher concentration specimens (0.05–0.5%) exhibited reduced individual droplet areas alongside detectable background fluorescence, indicating interfacial saturation by nanofibrils. The surplus nanofibrils were observed to form interconnected network architectures between adjacent droplets, which effectively suppressed coalescence and enhanced emulsion stability through the formation of physical barrier [49,50]. These microscopic observations provide direct visual evidence for the interfacial adsorption and network formation, offering qualitative support for the stabilization mechanisms discussed.

The CLSM-observed differences in interfacial adsorption behavior and droplet dispersion state among L-NCh, M-NCh, and S-NCh can be further elucidated by the particle adsorption energy theory. For chitin nanofibrils with the same chemical composition, the decrease in length (from L-NCh to S-NCh) leads to an increase in specific surface area, which enhances the interaction between nanofibrils and the oil–water interface and thus reduces the ΔG of adsorption [51]. Lower ΔG means that nanofibrils are more likely to adsorb at the interface and are less prone to desorption, which is consistent with the complete fluorescent coverage of droplets observed in CLSM images. In contrast, longer L-NCh has a smaller specific surface area and higher ΔG, resulting in relatively weaker interfacial adsorption capacity and a higher tendency to desorb from the interface, which may be one of the reasons for the pronounced droplet aggregation in L-NCh-stabilized emulsions. This correlation between nanofibril length, adsorption energy, and interfacial adsorption behavior is supported by relevant studies [52,53], which confirmed that the morphological parameters of nanofibrillar stabilizers significantly affect their adsorption energy at the interface and, thus, the microscopic structure and stability of Pickering emulsions.

### 3.8. Rheological Properties

The rheological properties of the Pickering emulsions were characterized using both oscillatory and steady-state measurements. The viscosity profiles of emulsions containing nanofibrils of different concentrations and length exhibited shear rate dependence (Figure 7A–C). All emulsions exhibited shear-thinning behavior, where viscosity decreased with increasing shear rate. This phenomenon can be attributed to the hydrodynamic alignment of fluid components along the flow field and the weakening of intermolecular interactions [54,55]. Higher nanofibril concentrations led to increased emulsion viscosity, likely due to reduced droplet size and enhanced interdroplet interactions [56]. At equivalent concentrations, emulsions containing higher-length nanofibrils showed greater viscosity. L-NCh tends to form entangled three-dimensional networks in the aqueous continuous phase, effectively trapping oil droplets and strengthening interdroplet bridging, thereby significantly enhancing flow resistance and improving viscosity and elastic modulus. In contrast, S-NCh, while unable to form stable entangled networks, exhibits higher specific surface area and stronger interfacial adsorption, enabling dense and irreversible adsorption at the oil–water interface to create a compact, rigid barrier that suppresses droplet coalescence and Ostwald ripening. These structural differences suggest that interfacial adsorption-dominated stabilization (S-NCh) can surpass network-entanglement-based stability (L-NCh) under certain conditions, consistent with reports that short nanofibrils with high specific surface area exhibit superior interfacial stabilization in Pickering emulsions [57,58]. The observed rheological responses serve as strong indirect indicators of the formation of a cohesive interfacial layer or a percolating network within the system, consistent with the proposed stabilization models.

The frequency-dependent viscoelastic behavior was characterized by dynamic oscillatory tests (Figure 7D–I). Both the storage (G′) and loss (G″) moduli exhibited amplification with increasing frequency. At low frequencies, the emulsions displayed fluid-dominant behavior (G″ > G′), which transitioned to solid-like characteristics at higher frequencies as G′ surpassed G″. This transition, indicative of shear-induced microstructural breakdown, demonstrates a reversible fluid–solid transition governed by the applied mechanical stress conditions [57].

## 4. Conclusions

This study systematically investigated the structure–property relationship of chitin nanofibrils (NCh) with tailored lengths and their efficacy in stabilizing Pickering emulsions. It was demonstrated that both nanofibril length and concentration critically governed emulsion stability, with the shortest variant (S-NCh) imparting superior performance as evidenced by a minimal creaming index, reduced droplet size, and a homogeneous microstructure. A critical stabilizer concentration of 0.05% was identified, below which instability occurred due to insufficient interfacial coverage. The stabilization mechanism was elucidated through rheological analysis, which revealed shear-thinning behavior and solid-like viscoelasticity at high frequencies, along with direct CLSM observations of nanofibril adsorption at the interface and the formation of a continuous network between droplets. This work establishes nanofibril length as a critical design parameter for chitin-based Pickering stabilizers. The conclusions are supported by a convergence of indirect macroscopic and microscopic evidence, strongly indicative of interfacial activity. Future research employing direct interfacial characterization techniques, such as interfacial rheology or cryogenic electron microscopy, will be essential to obtain molecular-level insights into the adsorption kinetics, packing structure, and mechanical strength of the nanofibril-laden interface.

Despite these insights, this work is limited to model emulsion systems and the performance of NCh under complex food matrices and realistic processing conditions remains to be evaluated. Further research should explore the interactions of NCh with other food components, their stability during storage and thermal processing, and their scalability for industrial production. In terms of practical application, the findings offer a rational design principle for biomass-derived colloidal stabilizers, supporting the potential use of length-tailored chitin nanofibrils in food products such as low-fat spreads, emulsion-based sauces, and functional beverage systems, where enhanced stability and clean-label attributes are desired.

## Figures and Tables

**Figure 1 foods-15-00076-f001:**
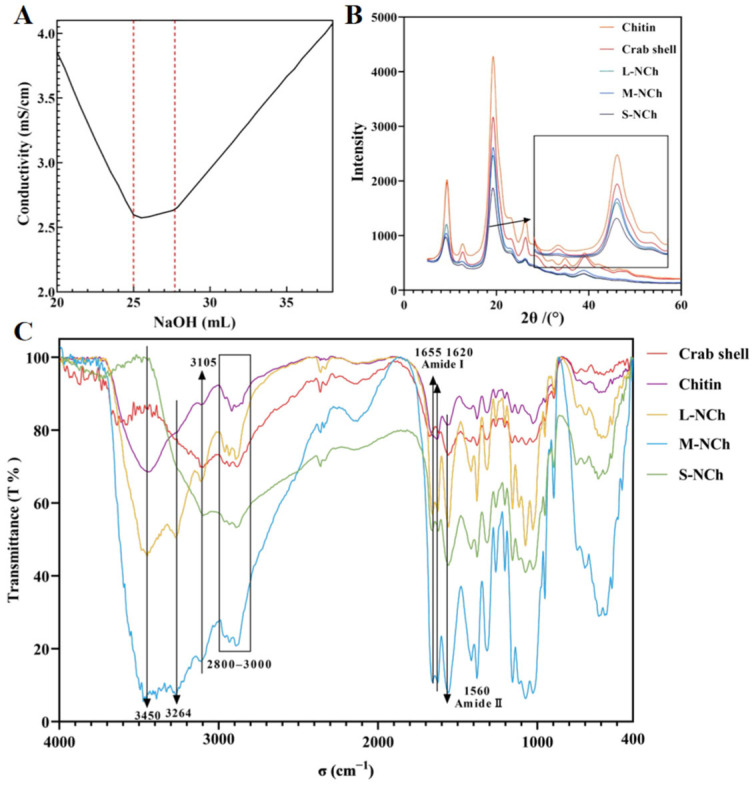
Deacetylation titration curve (**A**), X-ray diffraction patterns (**B**), and FTIR spectrum (**C**) of crab shell, chitin, L-NCh, M-NCh, and S-NCh. In (**A**), the two inflection points of the titration curve are marked by red dotted lines.

**Figure 2 foods-15-00076-f002:**
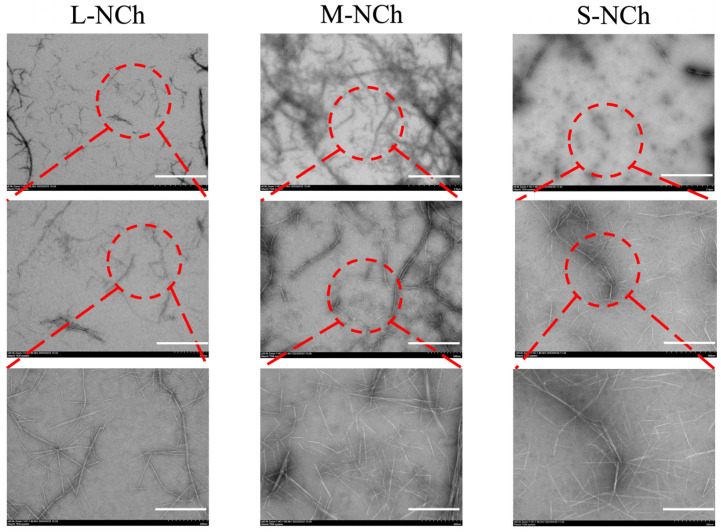
TEM analysis of L-NCh, M-NCh, and S-NCh samples. Red circles serve as visual guides, delineating the corresponding areas that are displayed at increasing magnifications across the image panels.

**Figure 3 foods-15-00076-f003:**
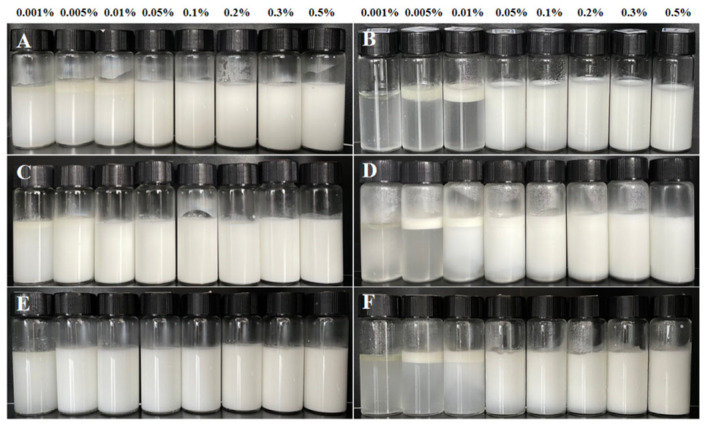
Appearance of Pickering emulsions stabilized by NCh with different lengths, immediately after preparation (Day 1) and after 7 days (Day 7). L-NCh Day1 (**A**), L-NCh Day7 (**B**), M-NCh Day1 (**C**), M-NCh Day7 (**D**), S-NCh Day1 (**E**), and S-NCh Day7 (**F**).

**Figure 4 foods-15-00076-f004:**
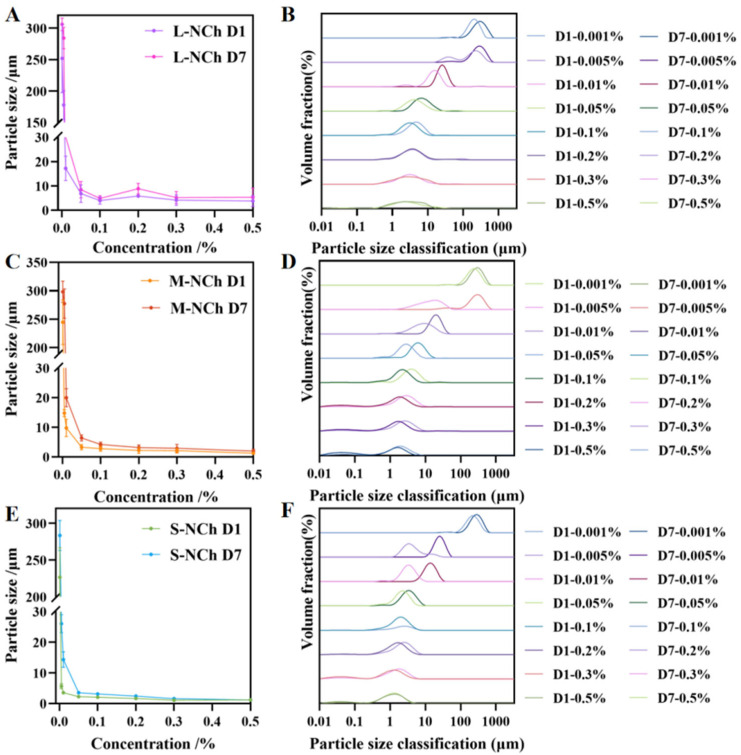
Particle size and distribution of Pickering emulsions stabilized with different NCh lengths ((**A**,**C**,**E**) represent the particle size of emulsion stabilized by L-NCh, M-NCh and S-NCh; (**B**,**D**,**F**) represent the particle size distribution of emulsion stabilized by L-NCh, M-NCh and S-NCh.).

**Figure 5 foods-15-00076-f005:**
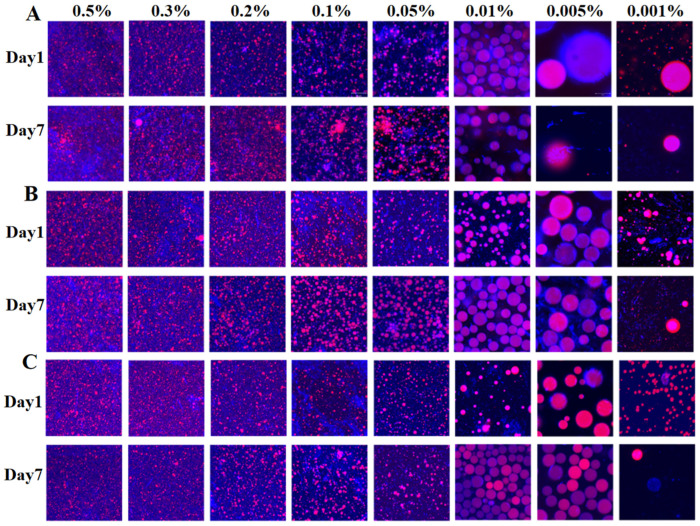
CLSM images of Pickering emulsions stabilized by NCh with different lengths on Day 1 and Day 7. (**A**) L-NCh, (**B**) L-NCh, and (**C**) S-NCh. Blue corresponds to chitin nanofibrils (NCh) in the aqueous phase; red denotes the oil phase; purple (in merged channels) indicates the interfacial region where NCh adsorbs onto oil droplets.

**Figure 6 foods-15-00076-f006:**
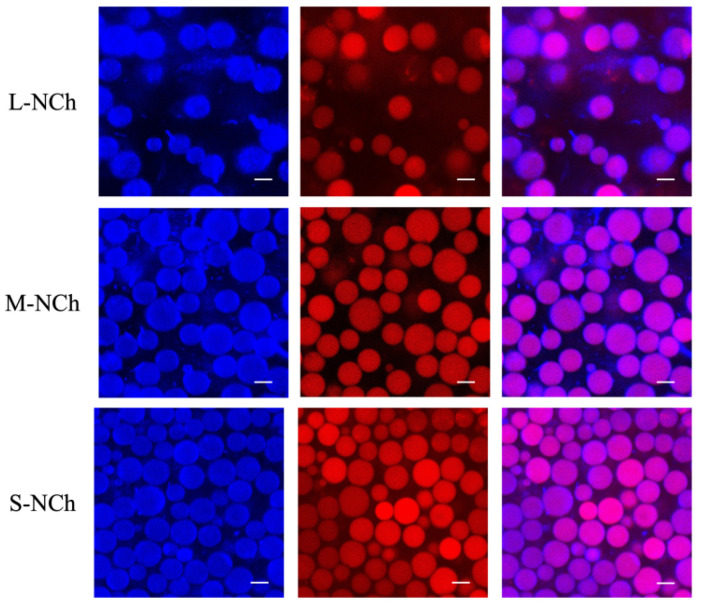
CLSM micrographs of Pickering emulsions by NCh with different lengths at a concentration of 0.3%. Blue corresponds to chitin nanofibrils (NCh) in the aqueous phase; red denotes the oil phase; purple (in merged channels) indicates the interfacial region where NCh adsorbs onto oil droplets.

**Figure 7 foods-15-00076-f007:**
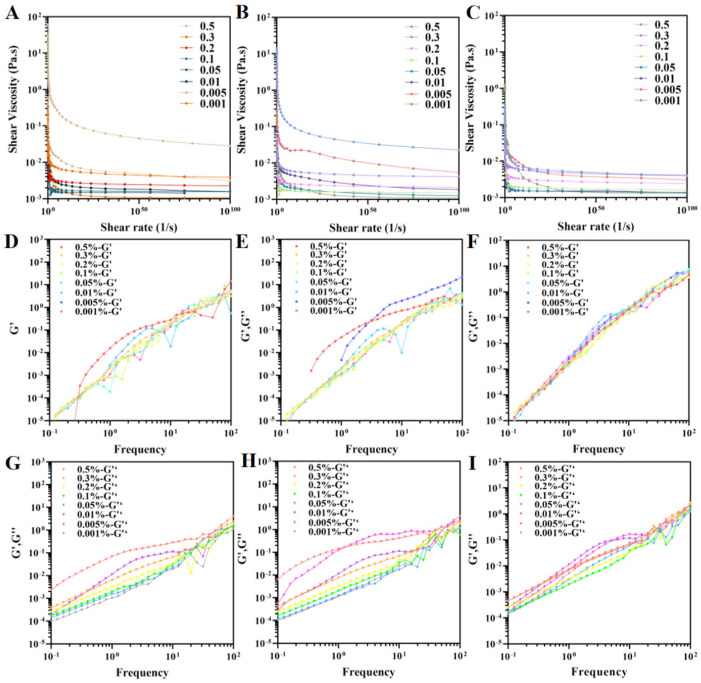
Rheological properties of Pickering emulsions stabilized by L-NCh, M-NCh, and S-NCh. Viscosity (**A**–**C**), storage modulus (G′) (**D**–**F**), and loss modulus (G″) (**G**–**I**) of Pickering emulsions stabilized L-NCh, M-NCh, and S-NCh, respectively.

**Table 1 foods-15-00076-t001:** Creaming index of Pickering emulsions stabilized by NCh with different lengths.

Nanofibril Concentration/%	Creaming Index/%		
	L-NCh	M-NCh	S-NCh
0.005	13.2 ± 5.06 ^c^	13.5 ± 3 ^d^	17.1 ± 0.03 ^a^
0.01	19.3 ± 1.31 ^c^	34.8 ± 16.55 ^c^	73.9 ± 13.44 ^b^
0.05	86.2 ± 3.32 ^ab^	87.1 ± 1.4 ^ab^	88.1 ± 0.02 ^ab^
0.1	87.6 ± 0.18 ^ab^	92 ± 3.87 ^a^	89.6 ± 1.2 ^ab^
0.2	92.8 ± 2.68 ^a^	92.3 ± 1.62 ^a^	94 ± 1.02 ^a^
0.3	95.7 ± 2.79 ^a^	94.9 ± 2.06 ^a^	95.9 ± 0.81 ^a^
0.5	100 ± 0 ^a^	100 ± 0 ^a^	100 ± 0 ^a^

Different superscript letters denote statistically significant differences (*p* < 0.05) in the creaming index among emulsions stabilized by the same length of NCh at different concentrations, whereas the same letter indicates no significant difference.

## Data Availability

The original contributions presented in the study are included in the article and Appendix A, further inquiries can be directed to the corresponding authors.

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
