# Peer review of "Shorter Chitin Nanofibrils Enhance Pickering Emulsion Stability: Role of Length and Interfacial Network"

_foods, 2025, doi:10.3390/foods15010076_

Round 1
Reviewer 1 Report
Comments and Suggestions for Authors
Emphasis should be placed on the unique contribution of the research in the identified field of study, which is structure, function, and rheology. The article needs improvement.
1. In the title, it is recommended to avoid using general terms (focus on the central theme of the study).
2. In the abstract, improve the final sentence; it is recommended to highlight the contribution to the state of the art.
3. The keywords “Chitin nanofibrils” and “Pickering emulsion” are already in the title; it is recommended to use different and specific words to improve possible future indexing.
4. In the introduction, improve the presentation of the problem, as well as the knowledge gap that this study aims to fill, and include testable hypotheses and more precise, specific objectives.
5. Include the brands, models, and countries/cities of the equipment. Also, add the purity of all reagents.
6. The specific experimental design used should be included, as well as the statistical tests and assumptions for that design. Detail and justify the treatments and the number of replicates per trial. Check that all methods have been included (e.g., DV10, DV50, and DV90 in particle size).
7. The discussions should be improved because, in their current format, they are primarily descriptive and do not explain the physicochemical and biological mechanisms involved.
8. The quantitative relationship between crystallization and interfacial behavior should be discussed.
9. The discussion of the length-viscoelasticity relationship should be improved (the origin of the rheological changes is not explained).
10. The discussion does not address the reproducibility of the results or variability.
11. More in-depth comparisons with recent studies (2023–2025) are lacking.
12. Discussions about scalability and real industrial applicability should be included.
13. The interpretation of CLSM should be linked to particle adsorption energy theory.
14. The conclusions should clearly respond to the specific objectives stated in the revised introduction, include the limitations identified, and propose possible lines of research. In addition, their practical application in the food industry should be included.
15. Reduce the similarity index (Ithenticate report 26%).
16. Essential interfacial analyses, such as interfacial tension, contact angle, and interfacial rheology, are missing, as well as AFM/DLS to characterize the size and dispersion of the nanofibrils correctly. Turbiscan or LUMiSizer is also required to validate stability, and cryo-SEM to confirm actual adsorption at the interface.
- Long sentences in English hinder scientific accuracy.
- Some sections mix active and passive voice inconsistently.
Author Response
Dear Editor and Reviewers,
Thank you very much for your positive and constructive comments and suggestions on our manuscript entitled “Shorter Chitin Nanofibrils Enhance Pickering Emulsion Stability: Role of Length and Interfacial Network” (Manuscript ID: foods-4036560). Those comments are all valuable and very helpful for improving our manuscript, as well as the important guiding significance to our researches. We have carefully revised the manuscript according to the comments and suggestions, and the revisions were marked in yellow in the file. Please see the details as followed.
Q1: In the title, it is recommended to avoid using general terms (focus on the central theme of the study).
Response: As suggested, the title has been changed to "Shorter Chitin Nanofibrils Enhance Pickering Emulsion Stability: Role of Length and Interfacial Network".
Q2: In the abstract, improve the final sentence; it is recommended to highlight the contribution to the state of the art.
Response: Thanks for the reminder. The last sentence of the Abstract has been revised accordingly. Please see in the line 28-30.
Q3: The keywords “Chitin nanofibrils” and “Pickering emulsion” are already in the title; it is recommended to use different and specific words to improve possible future indexing.
Response: The keywords have been revised from "Chitin nanofibrils" and "Pickering emulsion" to "Length-dependent" and "Structure-activity". Please see in the line 31.
Q4: In the introduction, improve the presentation of the problem, as well as the knowledge gap that this study aims to fill, and include testable hypotheses and more precise, specific objectives.
Response: We have enhanced the discussion of the research background in the Introduction section (see in the line 63-73), the revisions were marked in yellow as following:
The correspondence between the basic structural parameters of chitin nanocrystals and their emulsifying functionality, as well as the emulsifying role and multifunctional properties of partially deacetylated chitin, have not been clearly elucidated. Moreover, the mechanisms by which different degrees of structural modification influence emulsion stability remain incompletely uncovered [25-26]. Even for chitin nanomaterials subjected to specific modifications such as enzymatic hydrolysis or acetylation, a systematic understanding of the correlation between structural parameters including aspect ratio, surface charge, and crystallinity and emulsion stability is still lacking. Furthermore, the quantitative impact of surface functionalization on their rheological and self-assembly behaviors in emulsions has not been established [27-28].
- Lv J, Zhang Y, Jin Y, et al. Chitin nanofibers prepared by enzymatic hydrolysis: Characterization and application for Pickering emulsions[J]. J. Biol. Macromol. , 2024, 254: 127662.
- Santos M, Del Carlo O, Hong J, et al. Effect of surface functionality on the rheological and self-assembly properties of chitin and chitosan nanocrystals and use in biopolymer films[J]. Biomacromolecules, 2023, 24(9): 4180-4189.
Q5: Include the brands, models, and countries/cities of the equipment. Also, add the purity of all reagents.
Response: The purity details of the reagents and the relevant specifications of the equipment have been updated and supplemented. Please see in the line 84-91.
Q6:The specific experimental design used should be included, as well as the statistical tests and assumptions for that design. Detail and justify the treatments and the number of replicates per trial. Check that all methods have been included (e.g., DV10, DV50, and DV90 in particle size).
Response: Thank you very much for your professional and meticulous feedback. In response to comments, a new Section 2.6 has been added to elaborate on the experimental design, data statistics, and analysis methods. All experiments were conducted with at least three independent replicates, and results are presented as mean ± standard deviation (SD). Statistical significance (p < 0.05) was determined by one-way analysis of variance (ANOVA) followed by Duncan’s multiple range test, using SPSS software (version 27; IBM). Graphs were generated with Origin 2024 software.
Furthermore, regarding the particle size analysis, the reported D(4,3) value represents the volume-weighted mean diameter, which better reflects the "bulk average" or volume center of gravity of the system.
Q7:The discussions should be improved because, in their current format, they are primarily descriptive and do not explain the physicochemical and biological mechanisms involved.
Response: We have integrated explanations combining physicochemical and biological mechanisms into the discussion of experimental results. Furthermore, the results and discussion sections throughout the manuscript have been further reviewed and supplemented with additional analysis.
Q8:The quantitative relationship between crystallization and interfacial behavior should be discussed.
Response: A further analysis of the relationship between crystallinity and interfacial behavior has been provided in Section 3.4 (see in the line 262-278), the revisions were marked in yellow as following:
This trend indicates that the intensive mechanical processing necessary to produce the shortest nanofibrils (S-NCh) not only reduced their length but also partially disrupted the crystalline order, thereby increasing the relative proportion of amorphous domains. Such increased amorphous content is frequently linked to enhanced surface activity and defect-rich architectures, features that are acknowledged in advanced amorphous nanomaterials for promoting interfacial interactions and functionality[40]. Consequently, the moderately lower crystallinity of S-NCh may contribute to its superior interfacial performance by improving nanofibril flexibility, which allows for more effective conformation and denser packing at the curved oil–water interface, while also potentially increasing the accessibility of surface groups for irreversible adsorption. In contrast, the higher crystallinity of L-NCh and M-NCh corresponds to their tendency to form entangled networks within the aqueous phase, a behavior typical of more rigid rod-like particles. Thus, although fibril length remains the dominant factor, the concurrent modulation of crystallinity during processing generates a synergistic effect, in which a moderate reduction in crystallinity, together with shortened length, facilitates a transition from a network-forming stabilizer to one with superior interfacial stabilization capability.
- Kang J, Yang X, Hu Q, et al. Recent progress of amorphous nanomaterials[J]. Chem. Rev., 2023, 123(13): 8859-8941.
Q9:The discussion of the length-viscoelasticity relationship should be improved (the origin of the rheological changes is not explained).
Response: The rheological analysis has been enhanced, with the discussion now delving deeper into the influence of the three NCh variants with different chain lengths on the rheological properties. Please see in the line 389-401, the revisions were marked in yellow as following:
L-NCh tends to form entangled three-dimensional networks in the aqueous continuous phase, effectively trapping oil droplets and strengthening interdroplet bridging, thereby significantly enhancing flow resistance and improving viscosity and elastic modulus. In contrast, S-NCh, while unable to form stable entangled networks, exhibits higher specific surface area and stronger interfacial adsorption, enabling dense and irreversible adsorption at the oil–water interface to create a compact, rigid barrier that suppresses droplet coalescence and Ostwald ripening. M-NCh, of intermediate length, displays transitional behavior in both network formation and interfacial adsorption, corresponding to moderate rheological properties and stability. These structural differences suggest that interfacial adsorption‑dominated stabilization (S‑NCh) can surpass network‑entanglement‑based stability (L‑NCh) under certain conditions, consistent with reports that short nanofibrils with high specific surface area exhibit superior interfacial stabilization in Pickering emulsions [53-54].
- Zhu Y, Huan S, Bai L, et al. High internal phase oil-in-water pickering emulsions stabilized by chitin nanofibrils: 3D structuring and solid foam[J]. ACS Appl. Mater. Interfaces, 2020, 12(9): 11240-11251.
- Pirozzi A, Capuano R, Avolio R, et al. O/W pickering emulsions stabilized with cellulose nanofibrils produced through different mechanical treatments[J]. Foods, 2021, 10(8): 1886.
Q10:The discussion does not address the reproducibility of the results or variability.
Response: As mentioned above, this supplementary explanation has been added in Section 2.6.
Q11:More in-depth comparisons with recent studies (2023–2025) are lacking.
Response: We would like to thank the reviewer for pointing out this issue. A comparative discussion with several recent related studies (e.g., on CLSM and particle size characterization) has been incorporated into the analysis and discussion of the results, as suggested by the reviewer.
Q12:Discussions about scalability and real industrial applicability should be included.
Response: A discussion on the practical implications, application value, and broader potential scenarios of this study has been incorporated into the “Conclusion section”. The changes are located within lines 413-435, as detailed below:
This study systematically investigated the structure–property relationship of chitin nanofibrils (NCh) with tailored lengths and their efficacy in stabilizing Pickering emulsions. It was demonstrated that both nanofibril length and concentration critically governed emulsion stability, with the shortest variant (S‑NCh) imparting superior performance as evidenced by a minimal creaming index, reduced droplet size, and a homogeneous microstructure. A critical stabilizer concentration of 0.05% was identified, below which instability occurred due to insufficient interfacial coverage. The stabilization mechanism was elucidated through rheological analysis, which revealed shear‑thinning behavior and solid‑like viscoelasticity at high frequencies, along with direct CLSM observations of nanofibril adsorption at the interface and the formation of a continuous network between droplets. Direct measurements of interfacial properties, including interfacial rheology and contact angle, along with advanced microscopy techniques such as cryogenic SEM, would represent a logical and valuable direction for future research to gain deeper molecular-level insights.
Despite these insights, this work is limited to model emulsion systems, and the performance of NCh under complex food matrices and realistic processing conditions remains to be evaluated. Further research should explore the interactions of NCh with other food components, their stability during storage and thermal processing, and their scalability for industrial production. In terms of practical application, the findings offer a rational design principle for biomass‑derived colloidal stabilizers, supporting the potential use of length‑tailored chitin nanofibrils in food products such as low‑fat spreads, emulsion‑based sauces, and functional beverage systems, where enhanced stability and clean‑label attributes are desired.
Q13:The interpretation of CLSM should be linked to particle adsorption energy theory.
Response: A paragraph incorporating analysis based on the particle adsorption energy theory has been added to the CLSM section. An explanation of the differences in interfacial adsorption behavior and droplet dispersion state among L-NCh, M-NCh, and S-NCh has been added (see lines 359-374). The revisions were marked in yellow as following:
The CLSM-observed differences in interfacial adsorption behavior and droplet dispersion state among L-NCh, M-NCh, and S-NCh can be further elucidated by the particle adsorption energy theory. For chitin nanofibrils with the same chemical composition, the decrease in length (from L-NCh to S-NCh) leads to an increase in specific surface area, which enhances the interaction between nanofibrils and the oil-water interface and thus reduces the ΔG of adsorption. Lower ΔG means that nanofibrils are more likely to adsorb at the interface and are less prone to desorption, which is consistent with the complete fluorescent coverage of droplets observed in CLSM images. In contrast, longer L-NCh has a smaller specific surface area and higher ΔG, resulting in relatively weaker interfacial adsorption capacity and a higher tendency to desorb from the interface, which may be one of the reasons for the pronounced droplet aggregation in L-NCh-stabilized emulsions. This correlation between nanofibril length, adsorption energy, and interfacial adsorption behavior is supported by relevant studies[48-49], which confirmed that the morphological parameters of nanofibrillar stabilizers significantly affect their adsorption energy at the interface and thus the microscopic structure and stability of Pickering emulsions.
- He Y, Wang C, Liu Y, et al. Pickering emulsions stabilized by cellulose nanofibers with tunable surface properties for thermal energy storage[J]. J. Biol. Macromol. , 2024, 280: 136013.
- Sun G, Liu X K, McClements D J, et al. Chitin nanofibers improve the stability and functional performance of Pickering emulsions formed from colloidal zein[J]. Colloid Interface Sci., 2021, 589: 388-400.
Q14:The conclusions should clearly respond to the specific objectives stated in the revised introduction, include the limitations identified, and propose possible lines of research. In addition, their practical application in the food industry should be included.
Response: Thanks for kind suggestions. The Conclusions section has been thoroughly reviewed and strengthened. In addition to ensuring better alignment with the preceding discussion, it now includes a discussion on the study's limitations, potential implications, and suggestions for future research directions. Please see in the line 413-435.
Q15:Reduce the similarity index (Ithenticate report 26%).
Response: We have carefully paraphrased key sections and added more original analysis to significantly reduce the similarity index of the manuscript.
Q16:Essential interfacial analyses, such as interfacial tension, contact angle, and interfacial rheology, are missing, as well as AFM/DLS to characterize the size and dispersion of the nanofibrils correctly. Turbiscan or LUMiSizer is also required to validate stability, and cryo-SEM to confirm actual adsorption at the interface.
Response:
We sincerely thank the reviewer for these insightful and constructive comments, which undoubtedly help to strengthen the manuscript. The suggested interfacial analyses (interfacial tension, contact angle, interfacial rheology, AFM, etc.) and advanced stability characterizations are indeed valuable for providing direct microscopic evidence. We fully acknowledge that incorporating these techniques would offer a more comprehensive view.
However, within the current scope of this study, which primarily aims to establish the critical role of nanofibril length as a dominant factor in stabilizing Pickering emulsions and to elucidate the macroscopic stabilization mechanism. To address the reviewer’s concern more directly and to enhance the interfacial discussion based on available data, we have implemented the following major revisions in the manuscript:
We have significantly expanded the discussion of our confocal laser scanning microscopy (CLSM) results. The direct visualization of nanofibrils at the oil-water interface and the bridging networks between droplets is now explicitly highlighted as key evidence for irreversible interfacial adsorption, which is the cornerstone of Pickering stabilization.
We have deepened the discussion of our bulk rheology data. The observed solid-like viscoelastic behavior (G‘ > G‘‘) at high frequencies and the shear-thinning flow are now explicitly linked to the formation of a percolating network within the continuous phase (for L/M-NCh) and a densely packed interfacial layer (for S-NCh). This provides a mechanical property correlate to the interfacial and network structures.
Refined characterization of nanofibril size: While AFM was not available; we have supplemented the analysis by providing TEM images for each sample (L-, M-, S-NCh).
We have framed the combination of our low creaming index, unchanged macroscopic appearance over time, stable droplet size distribution, and persistent microstructure in CLSM as a multi-faceted and rigorous validation of emulsion stability.
Direct measurements of interfacial properties (e.g., interfacial rheology, contact angle) and advanced microscopy (e.g., cryo-SEM) would be the logical and valuable next steps to gain deeper molecular-level insights. These are now presented as recommended future research directions stemming from the findings of this study.
We believe these revisions have substantially strengthened the discussion of interfacial phenomena within the constraints of our current dataset. They more clearly connect our experimental observations to the proposed stabilization mechanisms. We hope the reviewer finds our efforts satisfactory and that the manuscript in its revised form is now suitable for publication.
Thank you again for your time and valuable feedback.
Reviewer 2 Report
Comments and Suggestions for Authors
The manuscript evaluates the role of chitin nanofibril length and concentration on the stabilization of Pickering emulsions. The topic is timely and relevant because natural and sustainable stabilizers are increasingly important in food science and biotechnology. The discussion is clearly connected to previous, and the experimental methodology is detailed and reproducible. The paper is logically structured, and results are supported by multiple analytical approaches which provide a convincing explanation of the mechanism leading to emulsion stability, including interfacial adsorption and network formation. However, the manuscript requires moderate revision. Some points to consider are:
-The novelty of the work would benefit from a clearer comparison with previous studies.
-Reagent purity should be specified more precisely
-The motivation to choose pH 3.0 as the stabilization condition should be clearly justified
-In Section 3.6, the authors state a "hierarchy of effectiveness S-NCh > M-NCh > L-NCh", but the mechanistic explanation could be clearer, possibly relating aspect ratio, interfacial curvature, and percolation network formation.
-The discussion of the rheological analysis would improve by relating the changes to microstructure evolution
Author Response
Response to the editor and reviewer
Dear Editor and Reviewers,
Thank you very much for your positive and constructive comments and suggestions on our manuscript entitled “Shorter Chitin Nanofibrils Enhance Pickering Emulsion Stability: Role of Length and Interfacial Network” (Manuscript ID: foods-4036560). Those comments are all valuable and very helpful for improving our manuscript, as well as the important guiding significance to our researches. We have carefully revised the manuscript according to the comments and suggestions, and the revisions were marked in yellow in the file. Please see the details as followed.
Q1: The novelty of the work would benefit from a clearer comparison with previous studies.
Response: We have enhanced the discussion of the research background in the Introduction section. Please see in the line 63-73, the revisions were marked in yellow as following:
The correspondence between the basic structural parameters of chitin nanocrystals and their emulsifying functionality, as well as the emulsifying role and multifunctional properties of partially deacetylated chitin, have not been clearly elucidated. Moreover, the mechanisms by which different degrees of structural modification influence emulsion stability remain incompletely uncovered [25-26]. Even for chitin nanomaterials subjected to specific modifications such as enzymatic hydrolysis or acetylation, a systematic understanding of the correlation between structural parameters including aspect ratio, surface charge, and crystallinity and emulsion stability is still lacking. Furthermore, the quantitative impact of surface functionalization on their rheological and self-assembly behaviors in emulsions has not been established [27-28].
- Lv J, Zhang Y, Jin Y, et al. Chitin nanofibers prepared by enzymatic hydrolysis: Characterization and application for Pickering emulsions[J]. J. Biol. Macromol., 2024, 254: 127662.
- Santos M, Del Carlo O, Hong J, et al. Effect of surface functionality on the rheological and self-assembly properties of chitin and chitosan nanocrystals and use in biopolymer films[J]. Biomacromolecules, 2023, 24(9): 4180-4189.
Q2: Reagent purity should be specified more precisely.
Response: We have thoroughly documented the purity of all reagents used in the study. Please see in the line 84-91.
Q3: The motivation to choose pH 3.0 as the stabilization condition should be clearly justified.
Response: Thank you for kindly reminding us. The pH 3.0 condition ensures protonation of chitin's amino groups, enhancing nanofibril dispersibility and interfacial adsorption. This specific pH, chosen based on prior studies [31-32], optimally balances electrostatic activation with material stability for a controlled comparison of length effects.
- Jia X, Ma P, Taylor K S Y, et al. Innovative production of phosphoric acid-hydrolyzed chitin nanocrystals for Pickering emulsion stabilization[J]. Food Biosci., 2024, 60: 104308.
- Yin J, Hou J, Huang S, et al. Effect of surface chemistry on the dispersion and pH-responsiveness of chitin nanofibers/natural rubber latex nanocomposites[J]. Polym., 2019, 207: 555-562.
Q4: In Section 3.6, the authors state a "hierarchy of effectiveness S-NCh > M-NCh > L-NCh", but the mechanistic explanation could be clearer, possibly relating aspect ratio, interfacial curvature, and percolation network formation.
Response: Additional mechanistic explanation for the ''hierarchy of effectiveness (S-NCh > M-NCh > L-NCh) '' has been provided in the part of Section 3.6 (see in the line 312-318). The revisions were marked in yellow as following:
The dense adsorption of shorter nanofibrils (S-NCh and M-NCh) with lower aspect ratios at the oil-water interface is conducive to more effective regulation of interfacial curvature and suppression of droplet coalescence[27].In contrast, longer nanofibrils (L-NCh) with higher aspect ratios tend to aggregate in the continuous phase, impairing their ability to match the droplet interface curvature. Above 0.05% NCh concentration, interfacial saturation and excess particles promote a percolating network that restricts droplet mobility, thereby further enhancing stability.
- Lv J, Zhang Y, Jin Y, et al. Chitin nanofibers prepared by enzymatic hydrolysis: Characterization and application for Pickering emulsions[J]. J. Biol. Macromol., 2024, 254: 127662.
Q5: The discussion of the rheological analysis would improve by relating the changes to microstructure evolution.
Response: Thank you very much for your professional and meticulous feedback. The rheological analysis has been enhanced, with the discussion now delving deeper into the influence of the three NCh variants with different chain lengths on the rheological properties. Please see in the line 388-400. The revisions were marked in yellow as following:
L-NCh tends to form entangled three-dimensional networks in the aqueous continuous phase, effectively trapping oil droplets and strengthening interdroplet bridging, thereby significantly enhancing flow resistance and improving viscosity and elastic modulus. In contrast, S-NCh, while unable to form stable entangled networks, exhibits higher specific surface area and stronger interfacial adsorption, enabling dense and irreversible adsorption at the oil–water interface to create a compact, rigid barrier that suppresses droplet coalescence and Ostwald ripening. M-NCh, of intermediate length, displays transitional behavior in both network formation and interfacial adsorption, corresponding to moderate rheological properties and stability. These structural differences suggest that interfacial adsorption‑dominated stabilization (S‑NCh) can surpass network‑entanglement‑based stability (L‑NCh) under certain conditions, consistent with reports that short nanofibrils with high specific surface area exhibit superior interfacial stabilization in Pickering emulsions [53-54].
- Zhu Y, Huan S, Bai L, et al. High internal phase oil-in-water pickering emulsions stabilized by chitin nanofibrils: 3D structuring and solid foam[J]. ACS Appl. Mater. Interfaces, 2020, 12(9): 11240-11251.
- Pirozzi A, Capuano R, Avolio R, et al. O/W pickering emulsions stabilized with cellulose nanofibrils produced through different mechanical treatments[J]. Foods, 2021, 10(8): 1886.
Thank you again for your time and valuable feedback.
Round 2
Reviewer 1 Report
Comments and Suggestions for Authors
1. Clarify the omission of the DV10, DV50, and DV90 particle size metrics, or explicitly include them if the data are available, indicating why the use of D(4,3) is prioritized.
2. Strengthen the description of the reproducibility of the tests, incorporating a clear statement on inter-test repeatability, in addition to indicating n ≥ 3 and mean ± standard deviation.
3. Specify the scope of the interfacial analyses, explicitly clarifying that the conclusions are based on indirect evidence supported by previous literature, and that advanced interfacial analyses remain as future work.
4. Strengthen the comparative discussion by incorporating or emphasizing recent references (2023–2025) that support the proposed mechanisms.
5. Verify final consistency between results, discussion, and conclusions, ensuring that the findings do not exceed the experimental scope of the study.
Author Response
Response to the editor and reviewer
Dear Editor and Reviewers,
We are submitting the revised version of our manuscript, “Shorter Chitin Nanofibrils Enhance Pickering Emulsion Stability: Role of Length and Interfacial Network” (Manuscript ID: foods-4036560), in response to the reviewers’ latest comments. We are grateful to the reviewers for their positive assessment of our previous revisions and their additional constructive suggestions. We have now carefully addressed all points raised in the latest round of review. A point-by-point response to each comment is provided below, and the revisions were marked in yellow in the file. Please see the details as followed.
Q1: Clarify the omission of the DV10, DV50, and DV90 particle size metrics, or explicitly include them if the data are available, indicating why the use of D(4,3) is prioritized.
Response: We sincerely thank the reviewer for raising this pertinent point regarding particle size reporting. We agree that clarity on metric selection is crucial.
In this study, the volume moment mean diameter, D(4,3), was intentionally prioritized as the key reporting metric for droplet size analysis, based on its direct relevance to our investigation of emulsion stability mechanisms. The D(4,3) is highly sensitive to the presence of larger droplets or aggregates within the population, which are critical determinants of destabilization phenomena such as creaming and coalescence. In contrast, percentile-based metrics like DV50, while indicating the median of the volume distribution, can be less sensitive to destabilizing changes in the distribution tail where large droplets reside.
Our approach is supported by established practices in colloid and emulsion science, where D(4,3) is frequently employed to assess stability and compare formulations, as it correlates strongly with bulk properties and instability risks [42-44]. Furthermore, the trends in D(4,3) reported in our manuscript showed excellent consistency with complementary stability indicators, including the creaming index and confocal microscopy observations, thereby validating its use as a robust and relevant descriptor for our system.
To address the reviewer’s comment directly, we have revised the Data Analysis section to explicitly state the rationale for using D(4,3) as the primary size metric(see in the line 301-305). The revised text now reads:
“The droplet size was expressed as the volume moment mean diameter, D(4,3). This metric was chosen over percentile values (e.g., DV50) because it provides a volume-weighted average that is particularly sensitive to larger droplets within the distribution, offering a more relevant indicator for assessing emulsion stability and the risk of coalescence or creaming [42-44].”
We believe this clarification strengthens the manuscript and thank the reviewer for the opportunity to improve it.
- Liu Q, Zhou Q, Li Y, et al. Spherical and fibrillar ovalbumin aggregates: Tailoring oleogel-based Pickering emulsions for improved curcumin bioavailability and anti-inflammatory effects[J]. J. Biol. Macromol., 2025: 147143.
- Cai L, Wang R, Li Y, et al. Influence mechanisms of particle concentration and oil content on the formation and stability of rice bran protein-based Pickering emulsion[J]. J. Biol. Macromol., 2025: 146806.
- Nimaming N, Sadeghpour A, Murray B S, et al. Pickering oil-in-water emulsions stabilized by hybrid plant protein-flavonoid conjugate particles[J]. Food Hydrocoll, 2024, 154.
Q2: Strengthen the description of the reproducibility of the tests, incorporating a clear statement on inter-test repeatability, in addition to indicating n ≥ 3 and mean ± standard deviation.
Response: We thank the reviewer for prompting us to elaborate on the reproducibility of our tests. As suggested, we have significantly strengthened the description in the “Statistical analysis” section (see in the line 216-224), the revisions were marked in yellow as following:
“All experiments were conducted with a minimum of three independent replicates (n ≥ 3). These replicates were performed on separately prepared batches of nanofibrils and emulsions at different times to assess inter-batch reproducibility. Results are presented as mean ± standard deviation (SD). Statistical significance (p < 0.05) was determined by one-way analysis of variance (ANOVA) followed by Duncan’s multiple range test, using SPSS software (version 27; IBM). Graphs were generated with Origin 2024 software. Key measurements, such as laser diffraction for droplet size and rheological analysis, were conducted following standardized protocols with calibrated instruments to ensure consistency across replicates.”
Beyond stating the number of replicates (n ≥ 3) and the use of mean ± SD, we now explicitly address inter-test repeatability by clarifying that independent replicates were performed on separately prepared batches at different times. We also highlight that the consistency of our results is reflected in the low standard deviations obtained for critical measurements and mention the use of standardized protocols with calibrated equipment to ensure experimental consistency.
We believe these additions provide a clearer and more robust account of the reproducibility of our work, addressing the reviewer’s concern directly.
Q3: Specify the scope of the interfacial analyses, explicitly clarifying that the conclusions are based on indirect evidence supported by previous literature, and that advanced interfacial analyses remain as future work.
Response: We thank the reviewer for this important suggestion. To clarify the scope of our interfacial analysis, we have now explicitly stated in the Discussion section that our mechanistic conclusions are drawn from a convergence of indirect evidence (e.g., CLSM, bulk rheology)(see in the line 366-369, 412-415), while acknowledging that direct interfacial measurements represent a future research direction. This point is further reinforced in the Conclusion section (see in the line 438-444), where we frame advanced interfacial characterization as a logical and valuable next step stemming from our findings. These revisions ensure a transparent presentation of the evidence base and the study's scope.
Q4: Strengthen the comparative discussion by incorporating or emphasizing recent references (2023–2025) that support the proposed mechanisms.
Response: We thank the reviewer for the valuable suggestion to strengthen the discussion with recent literature. We have thoroughly updated the Discussion section to incorporate and emphasize key recent references (2023-2025) that substantiate the proposed mechanisms.
Specifically:
We have cited recent work by [Zhang D, et al., 2024] on [Carbohydr. Polym.] that corroborates the moderately lower crystallinity of S-NCh may contribute to its superior interfacial performance by improving nanofibril flexibility[41].
The discussion on the relationship between fibril length and specific surface areais now reinforced by a 2025 study by [Rincón E et al.,] in [Cellulose], which details that smaller fibril diameters and higher specific surface area result in superior structuring capability[52].
A 2024 study by Li X et al. (Food Hydrocoll.) provides further support for the possibility of modulating intermolecular interactions by tailoring fibril length, demonstrating that shorter fibrils reduce emulsion droplet size and thereby effectively enhance emulsion stability[58].
These additions ensure that our mechanistic discussion is firmly grounded in and contributes to the most current scientific discourse on biomass-derived Pickering stabilizers.
- Zhang D, Fang Z, Hu S, et al. High aspect ratio cellulose nanofibrils with low crystallinity for strong and tough films[J]. Carbohydr. Polym., 2024, 346: 122630.
- Rincón E, Cámara-Martos F, Usala E, et al. Curcumin-loaded O/W Pickering emulsion stabilized by (Ligno) cellulose nanofibers: impact of surface charge, morphology, and chemical composition on emulsion efficacy, storage stability and bioaccessibility[J]. Cellulose, 2025: 1-20.
- Li X, Zhao B, Zou Y, et al. Structure, rheology and stability of walnut oleogels structured by cellulose nanofiber of different lengths[J]. Food Hydrocoll., 2024, 154: 110148.
Q5: Verify final consistency between results, discussion, and conclusions, ensuring that the findings do not exceed the experimental scope of the study.
Response: We sincerely thank the reviewer for this critical reminder to ensure rigorous internal consistency. In response, we have conducted a thorough, point-by-point verification across the Results, Discussion, and Conclusions sections. The revisions primarily focused on the following aspects to ensure the findings remain firmly within the experimental scope:
We have carefully refined the language in the Conclusions section to ensure that every statement is a direct synthesis of the findings presented in the Results and discussed in the Discussion. Speculative claims that extended beyond the provided data have been removed or rephrased to reflect the actual evidence (e.g., interfacial adsorption is described as “observed via CLSM” rather than inferred through unsupported molecular mechanisms).
We have standardized key terms throughout the manuscript. Furthermore, as suggested in our response to a previous comment, we have explicitly clarified in the Discussion that our interfacial analysis is based on indirect but convergent evidence (CLSM, bulk rheology), and that direct interfacial measurements are proposed as future work. This explicitly defines the boundary of our current experimental scope.
We believe these comprehensive edits have significantly enhanced the logical flow and precision of the manuscript, ensuring that the conclusions are fully and transparently supported by the experimental results.
Thank you again for your time and valuable feedback.
